# Validation of a Low-protein Semi-Quantitative Food Frequency Questionnaire

**DOI:** 10.3390/nu14081595

**Published:** 2022-04-12

**Authors:** Sharon Evans, Catherine Ashmore, Anne Daly, Richard Jackson, Alex Pinto, Anita MacDonald

**Affiliations:** 1Birmingham Women’s and Children’s Hospital NHS Foundation Trust, Birmingham B4 6NH, UK; catherine.ashmore@nhs.net (C.A.); a.daly3@nhs.net (A.D.); alex.pinto@nhs.net (A.P.); anita.macdonald@nhs.net (A.M.); 2Liverpool Clinical Trials Centre, University of Liverpool, Brownlow Hill, Liverpool L69 3GL, UK; r.j.jackson@liverpool.ac.uk

**Keywords:** phenylketonuria (PKU), dietary patterns, food frequency questionnaire, validation, reproducibility

## Abstract

Analysis of dietary patterns and their role in long-term health is limited in phenylketonuria (PKU). Food frequency questionnaires (FFQ) are commonly used to assess habitual intake. A semi-quantitative 89-item FFQ with a portion size photographic booklet was developed for children with PKU as a tool for collecting data on habitual intake of foods, food groups, energy and macronutrient intake. Twenty children with PKU aged 11–16 years, 30 parents of children with PKU aged 4–10 years, and 50 age/gender-matched control children were recruited. To test reproducibility, FFQs were completed twice with a mean interval of 5 weeks (range: 4–10). In order to test validity, FFQs were compared with five 24-h dietary recalls with a mean interval of 10 days (range: 6–18). Energy and macronutrient intake and quantity/week of individual food items were calculated and compared. There was good reproducibility for the FFQ with macronutrient correlations *r* > 0.6 and good validity data with most correlations *r* > 0.5. Bland–Altman plots for reproducibility and validity showed mean levels close to 0 and usually within 2 standard deviations. FFQ comparisons of PKU and control groups identified expected differences in % energy from macronutrients (PKU vs. control: carbohydrate 59% vs. 51%, fat 26% vs. 33%, protein 15% vs. 16%). This FFQ for PKU produced comparable data to repeated dietary recalls and is a valid tool for collecting data on habitual food and nutrient intake. It will be useful in assessing changes in dietary phenylalanine tolerance of new pharmacological treatments for PKU.

## 1. Introduction

Phenylketonuria (PKU) is a rare genetic condition, resulting in the failure to metabolise the amino acid phenylalanine, resulting in severe neurocognitive disability if untreated. It is managed with a low phenylalanine diet supplemented with a protein substitute (either phenylalanine-free L-amino acids or glycomacropeptide (GMP), typically with additional micronutrients), and special low-protein foods (SLPFs). The remaining diet consists of food starches, sugars, fruit and low-protein vegetables.

Dietary pattern analysis is increasingly used to examine food intake and the synergistic effect of food and nutrients [1,2], but this is unreported in PKU. Conceptually, dietary patterns provide a broad picture of food and nutrient consumption and may be more predictive of disease risk than individual foods or nutrients. In dietary pattern analysis, food consumption patterns are characterised by habitual intake [3]. With PKU, although much is known about the dietary prescription, little is known about what is consumed, including food preferences, range of meal choices and food patterns. Whilst it is assumed that patients eat plentiful amounts of low-phenylalanine fruit and vegetables, evidence suggests the converse position [4,5,6]. Furthermore, food neophobia appears to be more prevalent in children with PKU [5,7,8], and they appear reluctant to eat a wide range of fruits and vegetables [6].

Food frequency questionnaires (FFQ) are common tools used to measure dietary patterns. Respondents are given a list of foods and they describe how often each is eaten, e.g., how many times per day/per week/per month [9]. Compared with traditional dietary assessment methods, such as food diaries or 24-h recalls, FFQs require limited health professional time (both in data collection and analysis), low participant commitment, and may be completed by individuals with lower education or motivation [9,10]. They can be completed on paper or electronically in hospital clinics or the home environment. The results obtained by FFQs represent usual intakes over time and are suitable for ranking subjects into low, medium or high intake groups for individual foods or nutrients.

A FFQ should be tailored to each diet therapy. In PKU, portraying the full range of diverse foods permitted in a phenylalanine-restricted diet is challenging. This includes SLPFs and differing food types and quantities based on natural protein tolerance. Patients with classical phenotypes may tolerate only 3 g/day of natural protein (150 mg/day phenylalanine) but mild phenotypes tolerate ≥25 g/day (1250 mg/day phenylalanine), resulting in varying dependencies on SLPFs and protein substitutes. Pharmaceutical treatments, such as sapropterin dihydrochloride (BH4), may increase natural protein intake and the types of foods consumed in a subset of patients with PKU [11].

Ideally, a FFQ should contain no more than 100 commonly eaten foods grouped into sections, as only marginal gain is associated with more detailed questionnaires [12]. All FFQs should be validated to ensure that they measure what is intended and that they yield consistent results from repeated samples over time. This in turn improves the quality of the data collected and enables comparisons between studies using the same tool. There are different types of validity, meaning that a questionnaire is never fully validated but is valid for certain populations under specified conditions [13]. Validation of FFQs can be achieved in various ways and it is suggested that a combination of methods should be used to assess reproducibility and validity [14]. Checking that the questionnaire content is relevant and valid (content validity), that it can differentiate between different subject groups (construct/discriminative validity), that it produces reliable/reproducible results (reproducibility) and compares well with an existing standard (criterion validity), provides more credibility to the resulting data. Similarly, reporting that experts established questionnaire face validity, that the questionnaire was pilot tested on a subset of participants for understanding and relevance, and that appropriate statistical tests were used, also improve the integrity of the data [15].

Any FFQ designed for PKU should be validated by comparing it with a control group population to demonstrate that it is able to distinguish between the variations in macronutrient intake associated with the different food items eaten in a phenylalanine-restricted diet. Food intake will also vary according to the age, ethnic, social, educational, and economic background of the study population. Thus far, only one PKU-specific FFQ has been validated from the USA; 29 adults/adolescents were studied, and they compared the results of a FFQ with a 3-day food diary [10]. Whilst this study found good agreement between the different dietary methods and between repeated measures of the FFQ for protein intake, it was not validated in children, is likely to be specific to the USA population for food types and portion sizes, and it did not report on the validity of energy or other macronutrient intake such as carbohydrates, fat or fibre.

The aim of this study was to develop and validate a semi-quantitative FFQ for use in children with PKU, providing a tool that collects data on habitual intake for foods, food groups, energy and macronutrient intake, which can be utilised for dietary pattern and lifestyle analysis nationally.

## 2. Materials and Methods

### 2.1. Construct Validity (Ability to Differentiate between Different Subjects)—Study Subjects

Fifty children with PKU and 50 age and gender-matched healthy control children were recruited to test the FFQ for construct validity, which is the ability to differentiate between the dietary patterns for different groups. For children aged 4–10 years, data was completed by a parent/carer with assistance from an inherited metabolic disorder (IMD) dietitian; and for children aged 11–16 years, data was completed by children with assistance from the parent/carer and IMD dietitian. Inclusion criteria for subjects with PKU comprised the following: diagnosed by newborn screening; dietary treatment only (i.e., not prescribed sapropterin), supplemented with a prescribed free/low phenylalanine protein substitute from diagnosis and SLPFs; and no co-existing medical conditions, other special dietary requirements or intercurrent infection. All subjects with PKU were recruited from Birmingham Children’s Hospital over a 30-month period (2018–2020). For control subjects, inclusion criteria comprised the following: age (within 6 months) and gender-matched to subjects with PKU; and on a regular diet (special diets, including vegan, vegetarian, and dairy-free were excluded). Control subjects were recruited from siblings of other children with inherited metabolic disorders, friends, or family of Birmingham Children’s Hospital staff during the same time period as PKU subjects. The average nutrient and individual food intake from the 2 FFQs for each group were compared to establish construct validity.

### 2.2. Content Validity (Checked by Experts in the Field)—Food Frequency Questionnaire Development

Other UK and European PKU centres who were members of the SSIEM (Society for the Study of Inborn Errors of Metabolism) or BIMDG (British Inherited Metabolic Diseases Group) were invited to share their food frequency questionnaires. Five questionnaires were received (2 from England, 2 Scotland and 1 Germany). From these, a draft FFQ was developed for PKU and then adapted for control children (low-protein meal choices were matched with regular foods in the control FFQ, and SPLFs were also added to the PKU FFQ). The FFQs were reviewed by 5 IMD dietitians and piloted on 5 children with PKU and 5 control children to assess content validity. Following minor modifications, an 89-item PKU FFQ (+11 general dietary pattern questions) and a 69-item control FFQ (+5 general dietary pattern questions) were produced, including portion sizes for each item. The difference in the number of food items on the 2 questionnaires was a result of the additional SLPFs on the PKU version. General dietary pattern questions focused on meal frequency, missed meals, addition of salt to food, vitamin and mineral supplements and frequency of eating out (restaurants/cafes). The PKU FFQ also included 6 questions about quantity and frequency of protein substitute dosage and the number and amount of protein exchanges (the weight of food/drink that yields 1 g protein or 50 mg phenylalanine is one exchange).

### 2.3. Reproducibility—Food Frequency Questionnaire

The FFQ was completed at recruitment and at the end of the study (1–2 months apart) to test for reproducibility (test/retest reliability). This length of time was chosen in order to minimise changes over time but also to minimise recall of previous answers. The questionnaires were administered by one of four trained IMD dietitians using a standard script (see Appendix A). The same dietitian completed both questionnaires with each subject. For each food item on the FFQ, both the number of daily and weekly food portions consumed were recorded. Items consumed less than once a week were omitted.

### 2.4. Portion Size Booklet

Photographic portion size booklets were designed to accompany each FFQ, with pictures of the portion sizes specified in the questionnaire. Of 425 food photographs (captured from a range of distances and angles for each food), 65 were selected for the PKU FFQ and 58 for the control FFQ (24 foods were the same; 21 were the same but for the PKU FFQ were SLPFs instead of regular foods, e.g., low-protein pasta, bread, burgers; 8 were the same food but differed in portion sizes to show the amount that was equivalent to a 1 g protein/50 mg phenylalanine exchange; the remaining were diet specific foods, e.g., SLPFs with no regular comparative, or meat products with no low-protein comparative). Foods were prepared and presented on a plain white plate or bowl or transparent glass, on a neutral background and photographed immediately to maximise the aesthetic appearance of the food. Foods that were pre-packaged in standard portion sizes or were equivalent in size to another food did not have photographic representation. Portion sizes for all foods (including SLPFs) were described by weight (g), volume (mL) (if applicable), and household measurements, e.g., tablespoons, teaspoons, glasses, slices, packets/sachets, whole items, or a combination of these. Portion sizes for protein/phenylalanine containing exchange foods were usually described as the amount yielding 1 g protein or 50 mg/phenylalanine, often described as a weight or volume range, e.g., 50–70 g to allow for small variations in intake. Average portion sizes were generally used, these could then be multiplied up (e.g., double) or down (e.g., halved) if a larger or smaller portion size was consumed. For dietary analysis purposes, the smaller value was used to calculate nutrient intake.

### 2.5. FFQ Database

In order to analyse energy and macronutrient intake (protein, fat, carbohydrate and fibre) from the FFQ, food items were assigned a nutrient content based on composition data compiled from *McCance and Widdowson, The Composition of Foods* [16] supermarket nutrient analysis data (Tesco website accessed May 2017) and SLPF nutrient composition data from manufacturers. For each FFQ item, the nutrient analyses were selected from one or more of these sources and the nutrient contents were averaged to obtain a single value for each nutrient. These values were then entered in the *Nutritics* [17] software computer analysis program as ‘new foods’, including portion sizes, and data from the FFQs were then analysed using the items as entered in *Nutritics*. Data entry was completed by the same dietitian and cross-checked for accuracy by a second dietitian. Nutrient intakes for each of the 2 FFQs for each subject were obtained and converted into average daily intakes for energy (kJ), carbohydrate, protein, fat, dietary fibre, starch and sugars. The 2 FFQs were then compared to establish reproducibility, and the average of the 2 FFQs compared with the average of the 24-h recalls to establish criterion validity.

### 2.6. Criterion Validity (Comparison with an Existing Standard)—24-h Dietary Recalls

As a means of comparison and to test criterion validity (comparison with an existing standard), five 24-h dietary recalls were completed with subjects by one of 4 trained IMD dietitians experienced in taking diet diaries and using the same standard script for asking questions. All food and drink consumed the day before were recorded including type and quantity, and time of day consumed. All 5 dietary recalls for each subject were completed by the same dietitian during the same time period as the FFQs (over a 1–2-month period) and with a minimum of 5 days between each one including at least 1 weekend day and no more than 2 of the same weekdays, to ensure a representative intake (Figure 1).

Dietary recalls were all analysed by the same dietitian using the *Nutritics* [17] computer analysis program, and results were averaged to obtain an average daily intake. These were then compared with the average of the 2 FFQs to establish criterion validity.

### 2.7. Anthropometry

Body weight, height and BMI (body mass index) were measured and z-scores calculated at recruitment. Weight was measured to the nearest 10 g using Seca electronic scales; length was recorded to the nearest 1 mm using a Seca 213 portable stadiometer (Seca, Hamburg, Germany).

### 2.8. Statistics—Sample Size

Data from a previous study [5] in a subset of children with PKU suggested that for a high fat, high carbohydrate food such as potato fries, power to detect a clinically relevant difference in mean intake of 1.7 days/week (the difference between a PKU group mean, μ_1_, of 3.2 days/week and a control group mean, μ_2_, of 1.5 days/week). Assuming a common standard deviation (SD) of 2.08 and a two-sided significance level of 0.05, a sample size of 33 in each group will have a power of 90%. Sample size calculations were performed using nQuery Advisor.

### 2.9. Data Analysis

Analyses were performed to evaluate the differences between the FFQs for PKU and control groups, as well as between the FFQ and dietary recalls data using GraphPad Prism version 6.01 for Windows, GraphPad Software, La Jolla California, USA. Continuous data were summarised as median (IQR) and categorical data were summarised as frequencies of counts with associated percentages. The strengths of association between dietary components were estimated using Spearman’s correlation coefficients with Wilcoxon signed rank tests used to evaluate any differences between PKU and control groups. Comparisons of continuous data were performed using Wilcoxon sign rank for paired data and rank sum test for unpaired data. Categorical data were compared between groups using a Fisher test. Bland–Altman methods were used to assess the agreement between the FFQ and the 24-h dietary recall data. A *p*-value of 0.05 was used to determine statistical significance throughout.

### 2.10. Ethical Approval

This study was conducted according to the guidelines laid down in the Declaration of Helsinki, and a favourable ethical opinion was obtained from the London—Queens Square National Research Ethics Service (NRES) Committee (REC reference: 15/LO/1463 and IRAS ID: 185896). Written informed consent was obtained from the parent/carer of all subjects, and assent from children was obtained where appropriate, according to their level of understanding.

## 3. Results

### 3.1. Subjects

#### 3.1.1. PKU Group

Fifty children (24 male) with PKU, (mean age 9.3 years; range: 4–16 years) on a phenylalanine-restricted diet only, were recruited from one specialist PKU centre (Birmingham Children’s Hospital, UK). No changes were made to dietary intake during the study period. For 30 of the children aged 4–10 years, their mothers completed the questionnaires with a dietitian, and for the remaining 20 children, (aged 11–16 years) they self-completed the questionnaires with assistance from parents/carers and a dietitian.

#### 3.1.2. Control Group

Fifty control children were age (within 6 months) and gender matched to the subjects with PKU. Questionnaires were either completed by parents/carers or by teenagers, following the same criteria as in the PKU group.

### 3.2. Demographics and Anthropometry

Most children were white UK/European origin (n = 45 subjects with PKU and n = 47 controls), with the remaining being of either Asian (n = 3 PKU, n = 2 control) or mixed-race (n = 2 PKU, n = 1 control) origin. There was no significant difference between mean z-scores for BMI, weight or height between PKU and control groups (see Appendix A).

### 3.3. Meal Patterns—FFQ 1 vs. FFQ 2 (Reproducibility) vs. Dietary Recalls (Criterion Validity-Comparison with an Existing Standard)

The mean time between the two FFQs was 5 weeks (range: 4–10). Recalls were completed at a mean interval of 10 days (range: 6–18).

For meal patterns, there was little difference between the two FFQs, or between the FFQ and dietary recalls (Table 1). For the PKU group, there was a difference in the median number of meals that were consumed for FFQ 2 (4 meals/day) compared with FFQ 1 and the dietary recalls (5 meals/day). The PKU group varied across assessment methods in the percentage consuming mid-morning snacks, and this group was also less likely to consume a mid-morning snack compared with controls (FFQ 2 *p* = 0.0005). However, some children took their protein substitute at this time.

### 3.4. Protein Exchanges and Protein Substitute Intake

Dietary patterns related to intake of protein substitute and natural protein exchanges were comparable between repeated FFQs and between FFQ and dietary recall data (Table 2). The median daily number of 1 g protein exchanges (50 mg phenylalanine) from the 24-h dietary recalls was 0.5 g protein (25 mg phenylalanine) less than prescribed (5.0 g vs. 5.5 g). Dietary recall data showed that 70% (n = 35) of children with PKU were taking their daily number of 1 g protein exchanges to within 0.5 exchanges of prescribed amounts. Of the remaining 30%, 6% (n = 4) were allocated ≥ 10 g/day of protein and 18% (n = 9) were prescribed ≥5 g/day, indicating that these patients were less protein restricted.

### 3.5. Macronutrient Intake

#### 3.5.1. Reproducibility (A Measure of Whether the FFQ Produces the Same Results at Different Times)—FFQ 1 vs. FFQ 2

There was no significant difference between the two FFQs for PKU for any nutrients or between the two control FFQs except for protein and starch (*p* = 0.05) in control children, with FFQ 1 reporting values slightly higher than FFQ 2 (Table 3). Similarly, correlation *r* values all exceeded 0.5 for nutrients in both PKU and control groups, showing good correlation between FFQs taken at different intervals. Bland–Altman plots also demonstrated no clinically significant differences with mean levels close to 0 and homogeneous data mostly within the upper and lower levels of agreement (2 standard deviations—SD) (Figure 2). 

#### 3.5.2. Criterion Validity (Comparison with an Existing Standard)—FFQ vs. Dietary recalls

For the PKU group, there was a trend for the FFQ to report higher intakes of all nutrients compared to the dietary recalls (Table 3). In the control group, the same was observed except for energy, starch and fat. Nutrient correlations for the PKU group were close to or above 0.5 (r) except for fat. For the control group, correlations were less strong, ranging from 0.33 to 0.55. Conversely, most Wilcoxon *p* values for the PKU group were significant except for protein and starch, whilst in the control group fewer nutrients showed statistical differences, with only protein, fat and energy not showing a difference. However, from a clinical perspective, differences were not of relevance. For example, the difference in sugar intake between FFQ and dietary recalls for the PKU group was around 25 g or approximately 1 tablespoon per day, whilst the difference in fat intake was around 5 g or 1 teaspoon of fat per day. The Bland–Altman plots show homogeneous data with most values falling within the upper and lower levels of agreement (2 SD) and mean values close to 0 (Figure 3). The exceptions to this were sugar and fibre for the PKU group, and sugar for the control group.

#### 3.5.3. Construct Validity (Ability to Distinguish between Different Groups)—PKU FFQ vs. Control FFQ

As expected, and in agreement with previous research, due to the composition of a phenylalanine restricted diet there were significant differences in macronutrient intake between the PKU and control groups when using the FFQ (Table 3). The PKU group had significantly higher carbohydrate and starch intakes, and a higher percentage of energy from carbohydrate and a lower percentage of energy intake from fat compared to controls.

### 3.6. FFQ Individual Food Items

#### 3.6.1. Reproducibility (A Measure of Whether the FFQ Produces the Same Results at Different Times)—FFQ 1 vs. FFQ 2

Most food items for both PKU and controls showed good correlation between FFQ 1 and 2 (*r* > 0.40), demonstrating good reproducibility (see Appendix A). Foods with a lower correlation coefficient were usually consumed by fewer than 10 subjects. The exceptions for the PKU group were vegetarian gummy sweets, pasta sauce, dried fruit and regular biscuits. For the control group, exceptions were meat pie, meat curry and butter/margarine.

Similarly, for commonly eaten foods (> 10 subjects) there was no significant difference (*p* > 0.05) between FFQ 1 and 2 for most food items in either group. Exceptions in the PKU group included: corn/rice/oat-based breakfast cereal, sweet drinks and vegetables containing phenylalanine < 75 mg /100 g and those with >100 mg/100 g. Exceptions in the control group included the following: dairy desserts, wheat-based breakfast cereals, mayonnaise/dressings, pizza and crackers. No items with a low *r* value (<0.40) were significantly different (*p* < 0.05).

#### 3.6.2. Criterion Validity (Comparison with an Existing Standard)—FFQ vs. Dietary Recalls

There was a trend for the FFQ to report higher intakes compared with the dietary recalls for just over half the items (n = 54/89, 61% PKU; n = 35/69, 51% control) (See Appendix A). Similarly, for most foods, the FFQ reported more people consuming individual foods than the dietary recalls (FFQ 85%, n = 76/89 foods vs. recalls 15%, n = 13/79 foods for PKU; FFQ 83%, n = 57/69 foods vs. recalls 17%, n = 12/69 foods for controls).

Most food items for both PKU and control groups showed good correlation between the FFQ and the dietary recalls (*r* > 0.40), demonstrating satisfactory criterion validity except for items consumed less often (<10 subjects). The exceptions for the PKU group were as follows: vegetarian gummy sweets, chips, vegetarian burgers and low-protein biscuits. For the control group, exceptions were greater in number and similar to those that varied between the 2 FFQ: meat pie, meat curry and butter/margarine, in addition to chips, processed meats, ice cream, cheese, cake, pizza, pasta and chocolate.

Similarly, for commonly eaten foods (>10 subjects) there was no significant difference between the FFQ and dietary recalls for most food items in either group. Those that did tended to be different than the items that had low *r* values (<0.40). Exceptions in the PKU group included vegetarian gummy sweets, and in the control group cake, gummy sweets and table sauces—which had low *r* values (<0.40) and were significantly different (*p* < 0.05).

There were some commonly eaten foods (>10 subjects consuming) that showed significant differences in the mean g/week consumed between the FFQ and dietary recalls in both control and PKU groups. These included the following: boiled, mashed and jacket potato, table sauce, crisps and vegetables containing phenylalanine >75 mg /100 g. These foods tended to be considerably higher in the FFQ, except for crisps which were lower compared with the dietary recalls.

#### 3.6.3. Construct Validity (Ability to Distinguish between Different Groups)—PKU FFQ vs. Control FFQ

There were significant differences between the intake of PKU and control groups using the FFQ, particularly in the foods expected to be different (see Appendix A). This included higher protein foods that were consumed in greater quantities by controls: milk, cheese, soft cheese, dairy desserts, cream, wheat-based breakfast cereal, sandwich spreads, milk sauces, legumes, vegetables containing phenylalanine >75 mg/100 g, eggs, meat pies, meat curries, sugar-free drinks (usually containing aspartame), hot chocolate powder, nuts/seeds, and regular varieties of bread, bread rolls, pasta, pizza, biscuits, cakes, puddings, jelly, chocolate, gummy sweets and crackers. In addition, the higher carbohydrate/fat foods allowed since they are low protein/aspartame-free, were higher in the PKU group; these included: sweet spreads, mayonnaise/dressings, sweetened drinks (aspartame free), sugar, other sweets and butter, in addition to vegetarian varieties of foods such as burgers, pies and curries. Additionally, some foods commonly used as protein exchange foods were higher in the PKU group: tinned pasta, processed potato and potato or corn-based crisps.

## 4. Discussion

This is the first FFQ validated for children with PKU, with data suggesting that it is an effective, accurate and practical tool for estimating energy and macronutrient intake as an alternative method to dietary recalls. It identified dietary patterns, the quality of natural protein consumed, and adherence with protein prescription.

This FFQ demonstrated excellent reproducibility when administered at a mean time interval of 5 weeks. PKU group meal patterns were similar, and all nutrients showed good correlations (*r* > 0.6). The protein amounts in the PKU group had a correlation of 0.91, demonstrating that the FFQ reliably estimated usual intakes with similar accuracy to repeated 24-h dietary recalls. In addition, individual foods generally showed good correlation (*r* > 0.4) if they were commonly consumed items (eaten by >10 individuals). Discrepancies between FFQ 1 and 2 for individual foods may be explained by differences in interpretation between the two questionnaires or in participant memory of the types of food consumed at the various time points. For example, parents were sometimes ambivalent about the sugar content of drinks their children consumed, particularly if drinks were consumed at school/nursery or outside of the home. However, this could equally vary across the dietary recalls. Furthermore, some foods on the FFQ were rarely consumed (by <3 individuals). In order for a food itemisied on a FFQ to contribute to absolute intake or differentiate between individuals, it should be eaten regularly and by a significant number of the study population [9]. Therefore, some foods were removed from the study FFQ following analysis.

A minimum correlation coefficient of 0.3 to 0.4 has been suggested to detect associations when validating FFQs [9]. In this study, all nutrient correlation coefficients were above 0.5 for the comparison of the 2 FFQs in both groups, and above 0.4 for the comparison of FFQ and dietary recalls, except for fat in both groups and energy and protein for the control group only. Similar correlation results were shown in other validation studies [18,19,20,21,22,23].

Bland–Altman plots were used to display the stability and direction of the bias across levels of intake [19]. Agreement was considered reliable if the difference between the two measures for reproducibility (FFQ 1 vs. FFQ 2) or validity (FFQ vs. recalls) was within 2 standard deviations (SD) of the mean [10]; the mean was close to 0; and demonstrated homogeneous data. Expert consensus suggests a combination of correlation or regression statistical methods together with Bland–Altman analysis should be used to assess reproducibility and validity of a FFQ, rather than any one single method [14].

To be truly valid, reported dietary intake from any assessment method should not be significantly different to actual intake, however there are practical difficulties with measuring ‘absolute validity’; thus alternatively, ‘comparative validity’ (comparing with an alternative or ‘reference method’) is reported [24]. There is no gold standard method for recording dietary intake, all have limitations: weighed food records require a high level of subject commitment, adherence and understanding that would have excluded some recruits from this study; 3-day food diaries represent the current diet, rather than typical or usual intake over time; doubly labeled water, a more accurate method for comparison of energy intake, is expensive and requires specialised equipment that may be intimidating to children. Repeated dietary recalls have been previously used for validating FFQs [22,23,25]; whilst single day recalls do not account for day-to-day variability in food intake or episodically consumed foods [19], we chose to complete multiple 24-h recalls over a 4–10-week period to capture a more realistic picture of usual intake over time. This approach is supported by a systematic review of the validity of different dietary assessment methods compared with doubly labeled water, suggesting that multiple 24-h dietary recalls conducted over at least 3 days and using parents as proxy reporters was the most accurate method for children aged 4–11 years [25]. Neither the FFQ nor the multiple 24-h dietary recalls are likely to measure actual macronutrient intake with precision, as both are subject to recall bias; however, both methods produced a similar picture of intake.

Our FFQ designed for PKU demonstrated acceptable criterion validity when compared with the chosen reference standard, repeated 24-h dietary recalls. Total natural protein intake only varied by 0.5 g (25 mg phenylalanine) per day between methods, which is a very good correlation. There was some variability between assessment methods for the percentages of children reporting that they consumed food at mid meals; however, this is likely to be something that varies in individuals from day to day. Some individuals with PKU may also choose to consume their protein substitute in place of a snack between meals so as not to reduce appetite at main meals.

In keeping with other validation studies comparing FFQs with other methods [18,19,20,22,23] there was a tendency for the FFQ to report higher nutrient and individual food intakes than the dietary recalls. FFQs have been reported to overestimate dietary intake in children resulting from the use of adult portion sizes [10]; however, we overcame this by developing pictorial child-size portions. The main difference between the FFQ and dietary recalls was not so much the quantity of a food consumed, but less variation in the types of foods consumed for the dietary recalls. This reflects one of the limitations of dietary recalls in that they only capture recent intake rather than habitual food intake.

Consistent with previous studies looking at the macronutrient content of the PKU diet [26,27,28] our results demonstrated that the FFQ can differentiate the differences in macronutrient and individual food intake between children with PKU and children in the general population that would be expected. This substantiates good construct validity.

FFQs rely on recall over a longer assessment period than other methods and hence are associated with less accurate quantification. It is suggested that children under the age of 8 years may have difficulty recalling food intake, estimating portion size and conceptualizing frequency of food consumption [24,25]. The ability to cognitively self-report dietary intake accurately is commonly given as approximately 12 years [24,25]. Previous research has shown that when older children complete a FFQ, they receive less assistance from parents, and this can result in a greater number of inaccuracies [2]. There may also be anomalies in data (from both dietary assessment methods) for adolescents due to inaccurate self-reporting and the highly variable food patterns commonly seen in this age group [19]. In this study, children aged 11–16 years completed the FFQ and recalls themselves which may have led to misreporting, although parents were able to assist. Furthermore, children with PKU have a more repetitive food pattern, receive dietary education and are accustomed to measuring portion sizes and completing dietary assessments. Correlations for the FFQ compared with the dietary recalls were stronger for the PKU subjects than for controls, suggesting that PKU subjects or their parents may have had better dietary recall than the control group [14,29].

Completion of any dietary assessment method may draw participants’ attention to their diets [9], and there is also the risk of subjects responding in a way that demonstrates good adherence only in the presence of a dietitian. FFQs were administered by an IMD dietitian trained and experienced in dietary assessment rather than self-completion due to anticipated initial difficulties of comprehension and interpretation. As four different dietitians were involved in administration, there may have been some degree of inter-rater reliability. However, a standard script was used to administer questionnaires and food recalls to minimise this. It is anticipated that with repeated use, parents/carers and adolescents (>12 years) would be able to self-administer the FFQ independently. Recent studies have demonstrated that technology-assisted methods, such as an online FFQs, performed equally as well in estimating intakes as doubly labelled water and other methods [23,24]. As such, further analysis of this tool after regular use and with an online version may be warranted.

## 5. Conclusions

A FFQ can simplify dietary data collection in PKU, particularly if patients are familiar with the tool and can complete it electronically before clinic appointments. This low-protein FFQ designed for use in patients with PKU yielded comparable data to repeated dietary recalls, and can be validly used to collect data on usual food and nutrient intake in place of other dietary assessment methods. It will also enable assessment of the dietary patterns that may lead to lifestyle diseases, such as obesity in PKU, and in turn will facilitate tailored health messages to the PKU population that will help to reduce the incidence of health-related illness. It could also be particularly important in assessing the impact of dietary changes associated with pharmaceutical treatments in PKU. Further testing of an online version of the FFQ is warranted.

## Figures and Tables

**Figure 1 nutrients-14-01595-f001:**
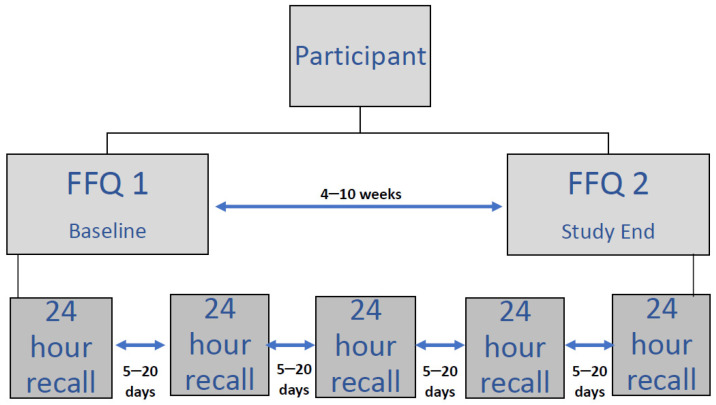
Study design. FFQ: Food frequency questionnaires.

**Figure 2 nutrients-14-01595-f002:**
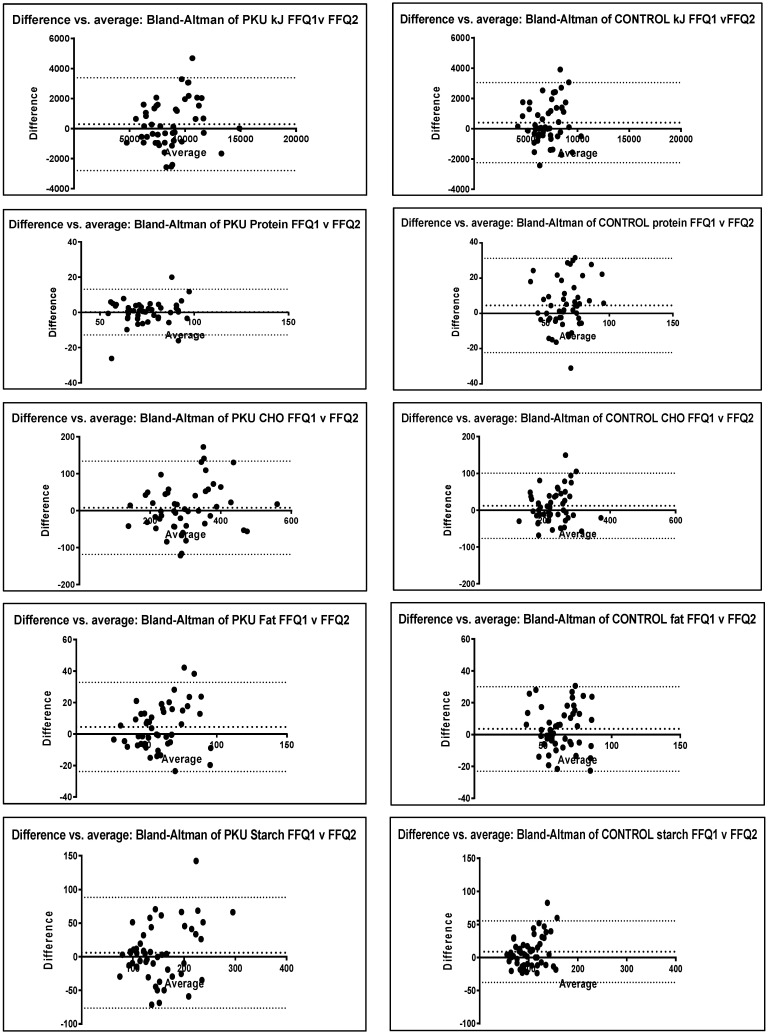
Bland–Altman plots for PKU and control group macronutrient intake FFQ 1 vs. FFQ 2. bias line (mean); upper and lower levels of agreement 95% confidence (2 SD). CHO = carbohydrate.

**Figure 3 nutrients-14-01595-f003:**
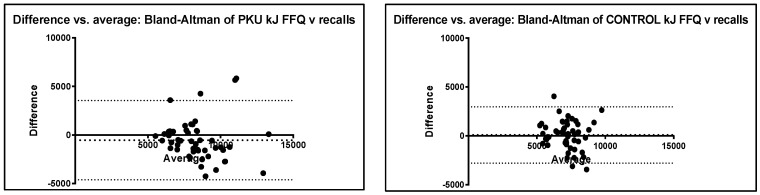
Bland–Altman plots for PKU and control group macronutrient intake FFQ vs. dietary recalls. Bias line (mean); upper and lower levels of agreement 95% confidence (2 SD); CHO = carbohydrate.

**Table 1 nutrients-14-01595-t001:** Meal patterns—percentage of subjects consuming meals and snacks, or missing meals for FFQ 1, FFQ 2 and dietary recalls (PKU and control groups).

	PKU				CONTROL			
	FFQ 1 n = 50	FFQ 2 n = 50	24-h Dietary Recalls ** n = 50	*p* Value	FFQ 1 n = 50	FFQ 2 n = 50	24-h Dietary Recalls ** n = 50	*p* Value
Median no. meals eaten/day	5	4	5	0.01 #	5	5	5	0.37 #
% eating breakfast (n)	92 (46)	94 (47)	100 (50)	1 *	100 (50)	100 (50)	100 (50)	1 *
% eating midday meal (n)	100 (50)	98 (49)	100 (50)	1 *	100 (50)	100 (50)	100 (50)	1 *
% eating evening meal (n)	100 (50)	100 (50)	100 (50)	1 *	100 (50)	100 (50)	100 (50)	1 *
% eating mid-morning snack (n)	62 (31)	38 (19)	48 (24)	0.027 *	78 (39)	74 (37)	60 (30)	0.815 *
% eating afternoon snack (n)	70 (35)	52 (26)	70 (35)	0.100 *	55 (28)	62 (31)	74 (37)	0.685 *
% eating bedtime snack (n)	40 (20)	46 (23)	52 (26)	0.686 *	46 (23)	54 (27)	58 (29)	0.549 *
% miss meals 1 x/week (n)	10 (5)	6 (3)	8 (4)	0.715 *	10 (5)	8 (4)	8 (4)	1 *
% miss meals > 1 x/week (n)	12 (6)	8 (4)	2 (1)	0.741 *	2 (1)	0 (0)	0 (0)	1 *

** average of 5-day dietary recalls; * Fisher test; # Wilcoxon signed rank; (n) = number of children; FFQ = Food Frequency Questionnaire; PKU = Phenylketonuria

**Table 2 nutrients-14-01595-t002:** Percentage of subjects consuming natural protein exchanges (1 g protein = 50 mg phenylalanine) and protein substitute at meals and mid meals (PKU group only).

	FFQ 1 n = 50	FFQ 2 n = 50	24-h Dietary Recalls ** n = 50	*p* Value
Median no. 1 g natural protein (50 mg phenylalanine) exchanges/day (range)	5.5 (3–25)	5.5 (3–25)	5.0 (2–23.5)	0.02 #
% eating prescribed protein exchanges at every meal (n)	62 (31)	60 (30)	66 (33)	1 *
% actually eating prescribed protein exchanges at every main meal (n)	74 (37)	60 (30)	46 (23)	0.202 *
Median no. meals/snacks per day that prescribed protein exchanges are consumed (range)	3 (1–6)	3 (1–6)	3 (1–5)	1 *
% eating prescribed protein exchanges at breakfast (n)	78 (39)	72 (36)	68 (34)	0.645 *
% eating prescribed protein exchanges at midday meal (n)	94 (47)	86 (43)	88 (44)	0.318 *
% eating prescribed protein exchanges at evening meal (n)	96 (48)	100 (50)	82 (41)	0.495 *
% eating prescribed protein exchanges at mid-morning snack (n)	4 (2)	2 (1)	4 (2)	1 *
% eating prescribed protein exchanges at mid-afternoon snack (n)	14 (7)	14 (7)	14 (7)	1 *
% eating prescribed protein exchanges at bedtime snack (n)	10 (5)	4 (2)	20 (10)	0.436 *
Median no. times/day protein substitute dose taken (range)	3 (3–5)	3 (3–5)	3 (3–5)	1 *
% taking protein substitute dose at breakfast (n)	100 (50)	100 (50)	100 (50)	1 *
% taking protein substitute dose at midday meal (n)	78 (39)	72 (36)	78 (39)	0.645 *
% taking protein substitute dose at evening meal (n)	74 (37)	68 (34)	62 (31)	0.660 *
% taking protein substitute with morning snack (n)	10 (5)	12 (6)	10 (5)	1 *
% taking protein substitute with afternoon snack (n)	36 (18)	34 (17)	26 (13)	1 *
% taking protein substitute with bedtime snack (n)	48 (24)	56 (28)	66 (33)	0.548 *

** average of 5 days; * Fisher test; (n) = number of children; # Wilcoxon signed rank.

**Table 3 nutrients-14-01595-t003:** Nutrients—median (IQR) intakes FFQ 1 vs. FFQ 2 and FFQ vs. Dietary Recalls (DR) for PKU and control groups; and FFQ PKU vs. control.

		PKU				CONTROL				Spearman Rank Correlation (*r*)				Wilcoxon Signed Rank (*p* value)				
		FFQ 1	FFQ 2	FFQ Average	Diet Recalls	FFQ 1	FFQ 2	FFQ Average	Diet Recalls	PKU FFQ 1 vs. 2	PKU FFQ vs. DR	Ctrl FFQ 1 vs. 2	Ctrl FFQ vs. DR	PKU FFQ 1 vs. 2	PKU FFQ vs. DR	Ctrl FFQ 1 vs. 2	Ctrl FFQ vs. DR	PKU FFQ vs. Ctrl FFQ
KJ/day	median	8116.8	8591.1	8259.7	7851.8	7011.7	6779.4	7144.2	7163.0	0.723	0.500	0.633	0.334	0.19	0.005	0.10	0.57	<0.0001
	(IQR)	(7078–11,288)	(6824–9446)	(7312–10,064)	(6806–8927)	(6008–8678)	(6274–7804)	(6157–8216)	(6318–7878)									
CHO g/day	median	277.3	293.7	287.8	267.1	226.9	215.4	221.0	208.0	0.682	0.601	0.651	0.408	0.62	0.01	0.17	0.02	<0.0001
	(IQR)	(229–394)	(237–343)	(241–356)	(231–315)	(191–271)	(194–247)	(186–262)	(183–233)									
Sugars g/day	median	135.7	128.8	131.2	107.0	112.2	111.6	115.7	95.7	0.78	0.464	0.719	0.550	0.78	0.0001	0.38	<0.0001	0.01
	(IQR)	(105–172)	(103–173)	(108–178)	(92–134)	(92–138)	(97–130)	(94–135)	(77–108)									
Starch g/day	median	136.3	146.8	144.2	140.5	102.8	96.7	100.1	116.8	0.731	0.600	0.742	0.493	0.51	0.25	0.05	0.004	<0.0001
	(IQR)	(114–183)	(105–187)	(114–195)	(117–167)	(81–132)	(79–115)	(82–124)	(93–133)									
Fat g/day	median	57.4	55.7	58.2	52.0	61.1	59.4	60.7	65.7	0.691	0.271	0.614	0.381	0.07	0.04	0.11	0.11	0.23
	(IQR)	(49–76)	(48–68)	(49–69)	(43–61)	(54–77)	(54–69)	(54–73)	(57–75)									
Total protein equivalent g/day	median	72.9	73.2	72.6	71.8	66.7	63.4	65.6	64.6	0.913	0.848	0.523	0.344	0.37	0.08	0.04	0.91	0.0009
	(IQR)	(65–84)	(66–82)	(65–83)	(68–80)	(55–77)	(55–73)	(57–74)	(56–76)									
Fibre g/day	median	20.4	22.6	21.5	18.1	18.0	17.4	17.8	17.0	0.673	0.445	0.514	0.427	0.70	0.001	0.26	0.02	0.0005
	(IQR)	(16–27)	(18–25)	(19–26)	(14–22)	(15–33)	(15–20)	(15–21)	(14–20)									
% Energy CHO	median	58.8	58.6	59.3	59.0	51.3	51.1	50.7	49.0	0.768	0.689	0.492	0.414	0.35	0.94	0.85	0.02	<0.0001
	(IQR)	(55–62)	(55–61)	(56–62)	(55–64)	(48–54)	(48–53)	(49–53)	(47–52)									
% Energy Fat	median	26.6	25.4	26.1	25.1	33.1	33.6	33.3	35.7	0.628	0.333	0.422	0.402	0.04	0.44	0.56	0.005	<0.0001
	(IQR)	(23–29)	(23–29)	(24–29)	(22–29)	(31–36)	(32–35)	(32–35)	(32–37)									
% Energy Protein	median	14.5	15.4	14.3	15.3	15.4	15.5	15.5	15.2	0.777	0.729	0.622	0.290	0.34	0.21	0.45	0.89	0.12
	(IQR)	(12–18)	(13–17)	(13–17)	(14–18)	(15–17)	(14–17)	(15–16)	(11–15)									

DR = Dietary recall, IQR = Interquartile range.

## Data Availability

Not applicable.

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
