# Peer review of "Validation of a Low-protein Semi-Quantitative Food Frequency Questionnaire"

_nutrients, 2022, doi:10.3390/nu14081595_

Round 1

Reviewer 1 Report

The paper present process of  developing  and validation of a semi-quantitative FFQ for use in children with phenylketonuria. The aim is very important taking into account the specificity od this disease, strictly connected with dietary restriction and specific food and protein exchanges intake. Nevertheless some comments have to be addressed

Abstract

Line 10: PKU - the abbreviation is not expanded and it is the first time when is used

Line 22:  CHO  the abbreviation is not expanded

Material and method

Lack of information where and when the children were recruited, especially control group.

Line 107: in author’s society, apart from children with vegan, vegetarian, and dairy-free diet,  there are only children with regular healthy diet? So it is the only population without children with so- called western diet?

Lack of information what time period was covered by FFQ

Looking at the FFQ preparation methodology - it's like two separate questionnaires, one for children with PKU, the other for healthy children (89 item PKU FFQ 118 (+11 general dietary pattern questions) and a 69-item control FFQ (+ 5 general dietary pat-119 tern questions) and only 24 food photographs in common), so it is rather obvious, that the differences will appear. I don't understand the idea of comparing two different tools to evaluate different eating habits to show that they are different.

Comments to the sample size calculation: it depends strictly on the purpose of the survey – did the authors want to find the clinically significant differences between PKU and healthy children or asses usual intake? And, in case of the first purpose, why potato fries are used as only determinant of those differences?

Data analysis:

Line 198: “ Direct comparisons between groups were assessed using the Spearman correlation coefficient. ' - sounds as if the authors used the Spearman correlation coefficient to compare consumption between PKU and the control group….

Line 201: “For longitudinal data 201 repeated measures of analysis of variance (ANOVA) or non-parametric equivalents (Friedman’s test) were applied – where? I didn't find this in the article

Results:

Line 236 – differences between PKU and control group was not tested - vide Table 1.

Line 242 : Table 1 – it does not represent the median intake

Line 253 – title – should include information that it is only for PKU group

Table 2.  please give the p -value of Wilcoxon signed rank test rather than NS. Many of other p-values in the table are also not significant, so I don’t see the reason why  the NS is used in this cases.

Moreover with median it should be IQR not range, or – to better understanding, first and third quartile. This is lacking also in Table 1.

Lines 249-252 – not clear, what was the idea?

Table 3 – kcal and kJ represent the same amount of energy – one is redundant in the table; not range but IQR should be given; mean with SD can be also be omitted, median with IQR much better represent the distribution of nutritional data

“FFQ v DR”  it is not clear what was compared. It should be clearly stated that it was average of the 2 FFQs (or at least I understand it in this way looking at the text earlier)   

Why were the tests not calculated for the structure of energy intake (% energy from fat, CHO, protein)?

In text below the table (no line numbers) –“ most t-test p values for the PKU group were significant” – there were not t-tests

Small were average differences, but level of agreement on Bland-Altman plots revealed the real discrepancies from mean from both tools, and in some cases they are quite big.

I suggest use also Kappa coefficient to assess the correct classification on the basis of FFQ to the high and low consumption of nutrients/food items.

Conclusions- to strong, in particular as to further use to the impact on dietary phenylalanine tolerance – it was not tested

Comments to the supplementary materials:

  • Table S1 – there is presented rather short characteristics, but not only BMI, weight and height, and please, standardize the presentation of data (delete y as years, “to/-“, give p not NS)

General comments – in my opinion at least one decimal place should be given when numbers represent a median and percentiles or percentages

Author Response

Please see attached response.

Reviewer 2 Report

 I was intrigued by the development of a FFQ that allows good estimation of eating patterns and quality of nutrition in food choices in PKU. It will be interesting to see how use of this FFQ allows comparison of nutritional intake between diet alone, sapropterin adjunct therapy, and pegvaliase enzyme substitution. The attention to detail and study design lend good validity to the instrument developed. My one suggestion would be to make the scripts used by the dietitians for the interviews available. I would find it important to know that questioning was non-directional. 

Author Response

Comments and Suggestions for Authors

 I was intrigued by the development of a FFQ that allows good estimation of eating patterns and quality of nutrition in food choices in PKU. It will be interesting to see how use of this FFQ allows comparison of nutritional intake between diet alone, sapropterin adjunct therapy, and pegvaliase enzyme substitution. The attention to detail and study design lend good validity to the instrument developed. My one suggestion would be to make the scripts used by the dietitians for the interviews available. I would find it important to know that questioning was non-directional. 

Thank you for your comments. The interviews have been made available as supplementary material.